# Spatial stratified heterogeneity analysis of field scale permafrost in Northeast China based on optimal parameters-based geographical detector

Ying Guo[1,2,3], Shuai Liu[1], Lisha Qiu[1], Chengcheng Zhang[1,2,3], Wei Shan[1,2,3]*

**1** Institute of Cold Regions Science and Engineering, Northeast Forestry University, Harbin, China, **2** Ministry of Education Observation and Research Station of Permafrost Geo-Environment System in Northeast China, Harbin, China, **3** Collaborative Innovation Centre for Permafrost Environment and Road Construction and Maintenance in Northeast China, Harbin, China

* shanwei@nefu.edu.cn

**Data Availability Statement:** All data used in the analysis are presented in figures and tables in the article. Raw data for this study should be obtained

## Abstract

Affected by global warming, the permafrost in Northeast China (NEC) has been continuously degrading in recent years. Many researchers have focused on the spatial and temporal distribution characteristics of permafrost in NEC, however, few studies have delved into the field scale. In this study, based on the Optimal Parameters-based Geographical Detector (OPGD) model and Receiver Operating Characteristic (ROC) test, the spatial stratified heterogeneity of permafrost distribution and the indicating performance of environmental variables on permafrost in NEC at the field scale were analyzed. Permafrost spatial distribution data were obtained from the Engineering Geological Investigation Reports (EGIR) of six highways located in NEC and a total of 19 environmental variables related to heat transfer, vegetation, soil, topography, moisture, and ecology were selected. The H-factors (variables with the highest contribution in factor detector results and interaction detector results): slope position (γ), surface frost number (SFN), elevation (DEM), topographic diversity (TD), and annual snow cover days (ASCD) were found to be the major contributors to the distribution of permafrost at the field scale. Among them, γ has the highest contribution and is a special explanatory variable for permafrost. In most cases, interaction can improve the impact of variables, especially the interaction between H-factors. The risk of permafrost decreases with the increase of TD, RN, and SBD, and increases with the increase of SFN. The performance of SFN to indicate permafrost distribution was found to be the best among all variables (AUC = 0.7063). There is spatial heterogeneity in the distribution of permafrost on highways in different spatial locations. This study summarized the numerical and spatial location between permafrost and different environmental variables at the field scale, and many results were found to be informative for environmental studies and engineering construction in NEC.

by contacting the Ministry of Education
Observation and Research Station of Permafrost
Geo-Environment System in Northeast China,
Harbin, Hexing Road 26, China (e-mail:
meors_pgsnec@163.com).

**Funding:** We thank the National Natural Science
Foundation of China (Grant No. 41641024) and
Science and the Technology Project of Heilongjiang
Communications Investment Group (Grant No.JT-
100000-ZC-FW-2021-0182) for providing financial
support and the Field scientific observation and
research station of the Ministry of Education-
Geological environment system of permafrost area
in Northeast China (MEORS-PGSNEC) for
providing original research data.

**Competing interests:** The authors have declared
that no competing interests exist.

## 1. Introduction

Permafrost is defined as ground that continuously remains at or below 0°C for at least two years [1]. Permafrost, as an important component of the cryosphere, is mainly distributed at high latitudes and altitudes in the Northern Hemisphere, and the area of permafrost in the Northern Hemisphere (permafrost probability >0) is approximately $19.82 \times 10^6$ km$^2$ [2]. Permafrost is related to ecology [3–5], hydrology [6, 7], carbon cycle [8, 9] and engineering [10, 11], and even public health [12]. Over the past decades, permafrost has been degraded worldwide [13–15], and the area occupied by the permafrost region decreased approximately from $22.79 \times 10^6$ km$^2$ [16] to $19.82 \times 10^6$ km$^2$ [2]. This change has caused widespread concern [17].

Northeast China (NEC) is one of the important ecological zones in the middle and high latitudes with rich vegetation resources and ecosystem services. However, relevant studies have shown that deforestation and grassland degradation in the region are serious problems, which have serious consequences for the ecosystem and human society [18, 19]. Ecosystems in such mid- and high-latitude areas of permafrost are considered to be more valuable for conservation [2]. Permafrost in NEC has been degraded over time and is one of the notable regions of permafrost degradation globally [3, 20, 21].

In the region of Da and Xiao Xing'anling Mountains (Greater and Lesser Khingan Mountains) in NEC, permafrost is known as the "Xing'an-Baikal" permafrost, This region exhibits distinct patterns of permafrost distribution, with permafrost being more extensively developed (colder, and thicker) at lower elevations in intermontane basins and lowlands, as opposed to elevational permafrost [22, 23]. Over the past few years, many researchers have focused on the distribution patterns and degradation trends of permafrost in NEC using various models, such as the Surface Freezing Number (SFN) model [21], TTOP model [24], GIPL model [25], and geographically weighted regression (GWR) model [26]. The common perception is that permafrost in NEC is degrading, and this degradation is related to topography, climate, vegetation, etc. The conclusions of these studies are based on data regarding the distribution of permafrost at a kilometer scale, with limited applicability at the field scale.

Permafrost conditions are usually highly variable at small spatial scales [27, 28]. Observations show that permafrost conditions (e.g., ALT) can be highly variable over short distances [29–32]. However, only few researches on permafrost have delved into the field scale in NEC. For instance, Wang et al. [33] generated high-resolution maps of permafrost distribution along the Bei'an-Heihe Expressway by intersecting surface temperatures from different seasons. Guo et al. [34] employed a maximum entropy classifier to produce a 30-meter resolution probability map of permafrost in Arxan. Despite the fine-scale nature of these studies, their methods and results exhibit strong specificity, making it challenging to extrapolate their findings to the entire NEC. Studying the spatial heterogeneity of permafrost at the field scale is an important work, but there are many limitations. Dense and systematic spatial observations of permafrost are scarce and many uninhabited sites are difficult to reach due to cost. Therefore, it is essential to analyze and summarize the patterns of permafrost distribution across the entire NEC at the field scale. During the construction of the highway, a large amount of geological survey data along the route will be obtained. These data are based on continuous spatial line segments. Although these data are not continuously observed on a time scale, these data can reflect the spatial heterogeneity of permafrost well.

Geographical phenomena commonly exhibit spatial heterogeneity, which is characterized by uneven distributions of geospatial attributes within a specific geographic area [35, 36]. To explore this heterogeneity across different strata of explanatory variables, spatial stratified heterogeneity analysis compares the spatial variance within each stratum with the variance between strata [37]. The Geographical detector (GD) model is a set of statistical methods to

study the spatial stratified heterogeneity of data, as well as to reveal the driving forces behind them [38], and its output $q$-values have a clear physical meaning. GD has no linearity assumptions and can objectively detect that the independent variables explain $100 \times q$% of the dependent variables. GD has been widely used for natural to social, such as land use [39], public health [40], economic change [41], ecology [42], and other field related to geospatial data analysis. Its study area is as large as a national scale and as small as a township scale. The Optimal Parameter-based Geographical Detector (OPGD) model was developed to be an enhanced version of the GD model that automatically selects the most suitable discretization method [37].

In this paper, we mainly analyze the spatial stratified heterogeneity of permafrost in NEC at the field scale by using the OPGD model with the Engineering Geological Investigation Reports (EGIR) of six highways and explore the contribution to permafrost from explanatory variables related to heat transfer, vegetation, soil, topography, moisture, and ecology. Through the factor detector and interaction detector of OPGD, we compare the differences in the contribution of explanatory variables for permafrost at the field scale. It is also hoped that the performance of variables in indicating permafrost could be derived by the risk detector of OPGD and Receiver Operating Characteristic (ROC) test. This study combines detailed soil and ground information to explore subtle variations in the distribution of permafrost, thereby providing a refined understanding and experience for field scale research on permafrost in NEC, and is expected to provide information and a reference to guide engineering and environmental practices at the field scale in NEC.

## 2. Data and methods

### 2.1 Data sources

Permafrost is a roadbed condition that requires special treatment in highway construction so its distribution must be clear. In the route selection and design stage of highway engineering, the geological conditions of the highway route need to be determined through drilling, and geophysical exploration, combined with preliminary information. After investigation, permafrost distribution data based on continuous spatial line segments along the route are obtained. In this study, only the distribution of permafrost, i.e. its presence or absence, was considered, without considering the properties of permafrost, such as permafrost temperature and active layer thickness (ALT). In other words, the distribution of permafrost is considered a 0–1 spatial distribution.

Taking NEC as the research area (Fig 1A), in this study, the data used are from Walagan-Xilinji section of national highway G111 (WX), Genhe-Mangui section of provincial highway S204 (GM), Kubuchun Forest Farm-Genhe section of national highway G332 (KG), Jiageda-Changqing Forest Farm section of national highway G331 (JC), Shiwei-Labudalin Highway section of national highway G331 (SL), and Yiershi-Chaiqiao section of provincial highway S308 (YC) (Fig 1C). Information about permafrost was extracted from EGIR of these highways (Table 1). The six highways are widely distributed in the permafrost region in NEC, and the permafrost temperatures at the highway locations range from 0˚C to -6˚C (Fig 1B). The total length of the six highways is 765.378 km, with a total of 260 permafrost sections and section length of 220.341 km. The EGIR for the SL is dated 2012, for the WX and YC is dated 2017, and for GM, JC, and KG is dated 2020. The GM highway has the longest permafrost section length of 112.975 km, and the KG highway has the highest proportion of permafrost length, accounting for 57.94%. Most of the highways are drilled to a maximum depth of 40 m. The YC highway has the highest elevation range of 872 to 1291 m.

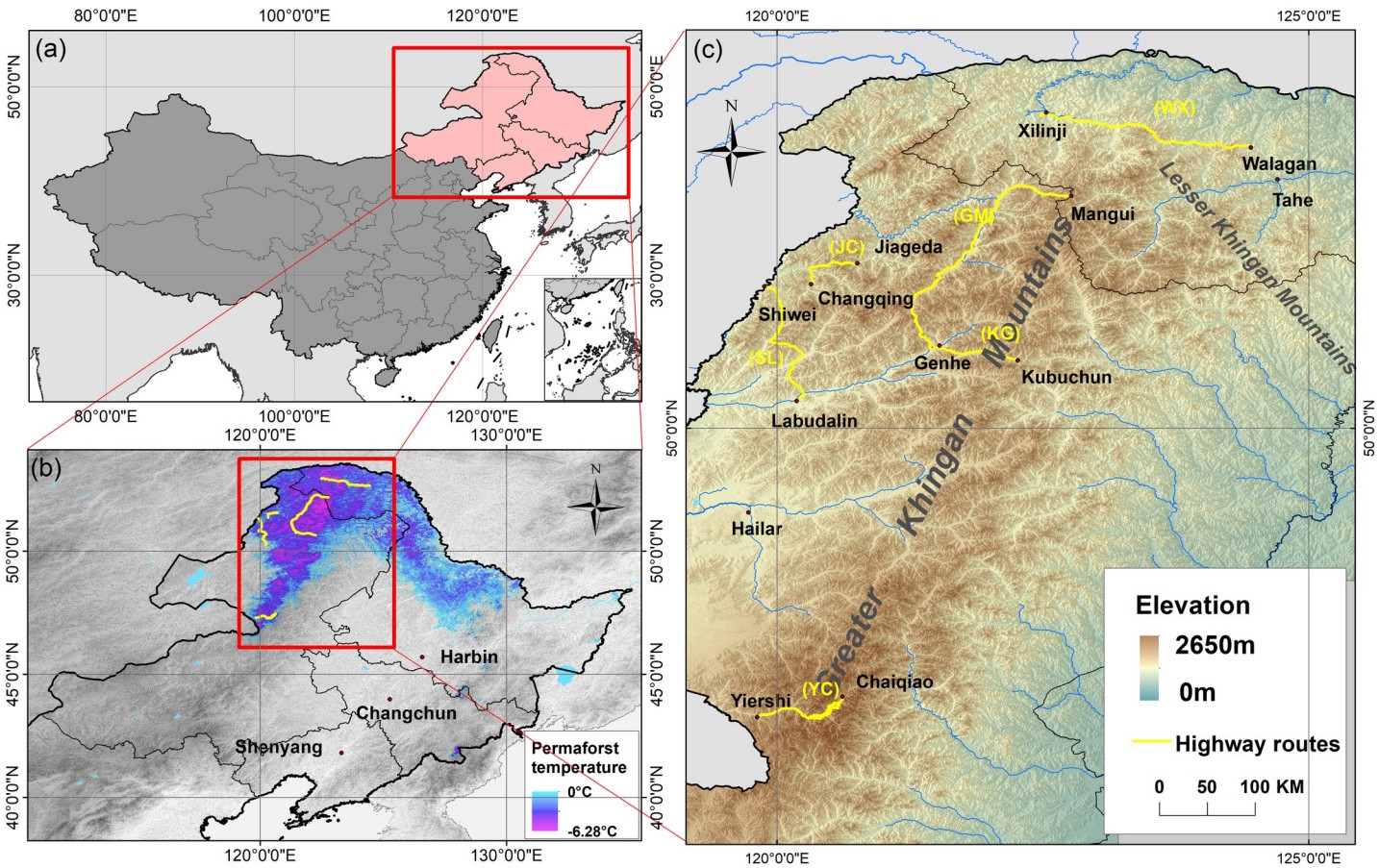

**Fig 1.** Spatial location of six highways (c) in the permafrost region (permafrost temperature < 0˚C, b) in Northeast China (a). The six highways are Walagan-Xilinji of national highway G111 (WX), Genhe-Mangui of provincial highway S204 (GM), Kubuchun forest farm-Genhe of national highway G332 (KG), Jiageda-Changqing forest farm of national highway G331 (JC), Shiwei-Labudalin of national highway G331 (SL) and Yiershi-Chaiqiao of provincial highway S308 (YC). Permafrost temperature data is provided by Shan et al. [43]. Elevation data was provided by NASA Land Processes Distributed Active Archive Center's free 30 m Shuttle Radar Topography Mission (SRTM) digital elevation model (DEM) ((SRTM V3, https://lpdaac.usgs.gov/, (accessed on 1 April 2023))). Boundary shapefiles (GS(2019)1822) is from the Ministry of Natural Resources of the People's Republic of China (https://www.mnr.gov.cn/, (accessed on 1 April 2023)).

## 2.2 Methods

GD model is a statistical methodology utilized to identify spatially stratified heterogeneity and uncover the underlying factors driving it. The central concept is founded on the assumption that when an independent variable significantly influences a dependent variable, their spatial distributions should exhibit similarity. GD can measure the spatial stratified heterogeneity of a given data; find the maximum spatial stratified heterogeneity of a variable; and find the explanatory variables of the dependent variable [36].

In this study, the OPGD model was applied in the analysis of permafrost distribution data. The OPGD-based analysis for this study consists of three parts: factor detector, interaction detector, and risk detector. In geographical studies, explanatory variables can take the form of both continuous and categorical variables. In the GD model, it is common practice to discretize and convert continuous variables into categorical variables. OPGD explores the optimal parameters for the best combination of spatial data discretization methods and breaks number of all explanatory variables counts for more accurate spatial analysis. It compares the q values derived from various discretization methods and chooses the one with the highest q value. In

**Table 1. Permafrost information of six highways.**

| No. | Highway name | Highway Code | Total length of highway (km) | Number of permafrost sections | Total length of permafrost section (m) | Number of holes drilled | Drilling depth (m) | EGIR report time | Highway longitude latitude range | Highway elevation range (m) |
|---|---|---|---|---|---|---|---|---|---|---|
| 1 | Walagan-Xilinji section of national highway G111 | WX | 157.012 | 129 | 47257 | 1112 | 3.5 to 40 | September 2017 | 122˚28'30"-124˚42'47"E 52˚21'33"-52˚57'17"N | 387–787 |
| 2 | Genhe-Mangui section of provincial highway S204 | GM | 255.000 | 48 | 112975 | 522 | 5 to 40 | May 2020 | 121˚15'25"-122˚46'15"E 50˚44'10"-52˚21'07"N | 608–1082 |
| 3 | Jiageda-Changqing Forest Farm section of national highway G331 | JC | 52.853 | 14 | 11520 | 161 | 5 to 25 | May 2020 | 120˚18'48"-120˚45'06"E 51˚21'34"-51˚33'38"N | 526–892 |
| 4 | Kubuchun Forest Farm-Genhe section of national highway G332 | KG | 59.690 | 34 | 34583 | 382 | 5 to 35 | December 2020 | 122˚13'37"-122˚46'15"E 50˚38'51"-50˚45'27"N | 703–950 |
| 5 | Shiwei-Labudalin Highway section of national highway G331 | SL | 153.744 | 33 | 29010 | 425 | 5 to 40 | July 2012 | 119˚52'47"-120˚15'40"E 50˚14'51"-51˚20'08"N | 481–900 |
| 6 | Yiershi-Chaiqiao section of provincial highway S308 | YC | 87.079 | 21 | 5351 | 243 | 5 to 40 | July 2017 | 119˚48'43"-120˚37'01"E 47˚16'00"-47˚28'12"N | 872–1291 |

this study, the alternative discretization methods used are equal breaks, natural breaks, quantile breaks, and geometric breaks. The number of breaks ranges from 5 to 13.

**2.2.1 Factor detector.** As the core part of GD model, the factor detector was used to identify the contribution of a single explanatory variable on permafrost distribution with a $q$-statistic. The $q$-statistic compares the dispersion variances between observations in the whole study area and strata of the spatial distribution of variables [36–38]. The $q$ value of an explanatory variable is computed by:

$$q = 1 - \frac{SSW}{SST} \tag{1}$$

$$SSW = \sum_{h=1}^{L} N_h \sigma_h^2, SST = N\sigma^2 \tag{2}$$

where $q$ means the explanatory variable explains $100 \times q\%$ of permafrost; $h$ ($h = 1,...,L$) is the strata of the explanatory variable and permafrost; $N_h$ and $N$ are the number of cells in the $h$th strata and the number of cells in the whole area; $\sigma_h^2$ and $\sigma^2$ represent the variance of the observations in the $h$th strata and the whole area. A large $q$ value means the relatively high importance of the explanatory variable, due to a small variance within strata and a large variance between strata; $SSW$ and $SST$ are the sum of variance within strata and the total variance of the whole area, respectively.

The $F$-test is utilized to determine whether the variances of observations and stratified observations are significantly different, since the transformed $q$ value can be tested with the

non-central $F$-distribution:

$$F = \frac{N - L}{L - 1} \frac{q}{1 - q} \sim F(L - 1, N - L; \delta) \tag{3}$$

$$\delta = \left[ \sum_{h=1}^{L} \bar{Y}_h^2 - \frac{1}{N} \left( \sum_{h=1}^{L} \bar{Y}_h \sqrt{N_h} \right)^2 \right] / \sigma^2 \tag{4}$$

where $\delta$ is the non-central parameter; $\bar{Y}_h$ is the mean value of observations within the $h$th strata. Thus, with the given significant level, the null hypothesis $H_0:\sigma^2 = \sigma_h^2$ can be tested by checking $F(L\text{-}1, N\text{-}L; \delta)$ in the distribution table.

**2.2.2 Interaction detector.** The interaction detector determines the interaction impacts between two variables by comparing the $q$ value of a single variable and the $q$ value of a two-variable interaction. The method is advantageous because the hypothesis of interaction is not limited to traditional statistical methods. The interaction of the explanatory variables is identified by the $q(X_i \cap X_j)$ value of the detector result to determine whether the interaction between the explanatory variables increases or decreases the impacts of permafrost. The interaction detector explored five types of interactions, including nonlinear weaken, uni-variable weaken, bi-variable enhance, independent, and nonlinear enhance [36, 38] (Table 2). Therefore, the results of the interaction detector include the $q$ value of the interaction and the type of interaction.

**2.2.3 Risk detector.** In the spatial stratified heterogeneity analysis, the risk detector is employed to assess the statistical significance of variations in spatial patterns represented by mean values across different strata. Specifically, the $t$-test is used to examine the difference between the mean values of strata $\eta$ and $k$. This test allows for evaluating the significance of variations in the spatial patterns among the different strata [36–38]:

$$t_{\bar{Y}_\eta - \bar{Y}_\kappa} = \frac{\bar{Y}_\eta - \bar{Y}_\kappa}{\left( \frac{s_\eta^2}{N_\eta} + \frac{s_k^2}{N_\kappa} \right)^{\frac{1}{2}}} \tag{5}$$

where $\bar{Y}_\eta$ and $\bar{Y}_\kappa$ are mean values of observations within strata $\eta$ and $k$, $s_\eta^2$ and $s_k^2$ are the sample variance, and $N_\eta$ and $N_k$ are numbers of observations. respectively. The statistic is

**Table 2. Interactions between two explanatory variables.**

| Geographical interaction relationship | Interaction |
|---|---|
| $q(A \cap B) < \min(q(A), q(B))$ | Nonlinear weaken |
| $\min(q(A), q(B)) < q(A \cap B) < \max(q(A), q(B))$ | Uni-variable weaken |
| $\max(q(A), q(B)) < q(A \cap B) < q(A) + q(B)$ | Bi-variable enhance |
| $q(A \cap B) = q(A) + q(B)$ | Independent |
| $q(A \cap B) > q(A) + q(B)$ | Nonlinear-enhance |

*$q(A)$ is the $q$ value of variable A, $q(B)$ is the $q$ value of variable B, and $q(A \cap B)$ is the $q$ value of the interaction between variables A and B.

approximately distributed as student's $t$ with the degree of freedom of:

$$df = \left(\frac{s_\eta^2}{N_\eta} + \frac{s_\kappa^2}{N_\kappa}\right) / \left[\frac{1}{(N_\eta - 1)}\left(\frac{s_\eta^2}{N_\eta}\right)^2 + \frac{1}{(N_\kappa - 1)}\left(\frac{s_\kappa^2}{N_\kappa}\right)^2\right] \tag{6}$$

Thus, with a given significant level, the null hypothesis $H_0 : \bar{Y}_\eta = \bar{Y}_\kappa$ can be tested with the student's $t$ distribution table.

**2.2.4 The Receiver Operating Characteristic (ROC) test.** The Receiver Operating Characteristic (ROC) test is a statistical analysis method used to evaluate the prediction performance of binary classifiers. The ROC curve uses the true positive rate (the ratio of present and predicted present) as the ordinate and the false positive rate (the ratio of not present but predicted present) as the abscissa. The area under the curve (AUC) was used to evaluate the performance of the indication. The larger the AUC value, the better the indication performance of the variable. If the indication accuracy is 100%, then AUC = 1; if the indication performance is a completely random indication, then AUC = 0.5 [44, 45].

## 2.3 Explanatory variables selecting and processing

Many environmental variables affect the occurrence of permafrost, such as heat transfer, vegetation, soil, terrain, moisture, and ecology. In this study, 19 environmental variables were selected as explanatory variables from these six aspects (Table 3). SFN and annual snow cover days (ASCD) were selected as variables related to heat transfer. SFN was calculated using MODIS daily land surface temperature (LST) data (MOD11A1 V6.1, https://lpdaac.usgs.gov/, (accessed on 1 July 2021)) with a resolution of 1km [43]. The SFN used in this study is the average annual value of SFN from 2014 to 2019. ASCD is calculated using MODIS daily snow cover data (MOD10A1.006, https://lpdaac.usgs.gov/, (accessed on 1 April 2023)) with a resolution of 500m. The ASCD used in this study are the average ASCD values for the 10 years prior to the EGIR reporting time of each highway.

Regarding the environmental variables of vegetation, based on the Normalized Difference Vegetation Index (NDVI) data provided by MODIS (MOD13Q1.061, https://lpdaac.usgs.gov/ (accessed on 1 April 2023)), the maximum NDVI (NDVImax) and mean NDVI (NDVImean) for the three years prior to the EGIR reporting time of each highway were selected to reflect the impact of vegetation. In addition, the vegetation type provided by the National Cryosphere Desert Data Center (http://www.ncdc.ac.cn, (accessed on 1 April 2023)) with a resolution of 1km was considered.

Among many soil properties, Soil Organic Carbon (SOC), Soil Coarse Fragments (SCF), and Soil Bulk Density (SBD) from SoilGrids 2.0 (https://soilgrids.org/, (accessed on 1 April 2023)) [46] were selected. They are taken as average values of 0-200cm below the surface.

In topographic analysis, many variables can be obtained based on the Digital Elevation Model (DEM). In this study, DEM data (SRTM V3, https://lpdaac.usgs.gov/, (accessed on 1 April 2023)) was provided by NASA Land Processes Distributed Active Archive Center's free 30m Shuttle Radar Topography Mission.

Topographic position index (TPI) and deviation from mean elevation (DEV) are two variables that can indicate how much a central point is located higher or lower than its average surroundings. TPI measures the difference between elevation at the central point ($z_0$) and the average elevation around it within a predetermined radius ($R$); DEV measures the topographic

**Table 3. Explanatory variables used in OPGD.**

| No. | Variable category | | Variable abbreviation | Variable type | Unit | Variable description | Data sources and references | Spatial resolution |
|---|---|---|---|---|---|---|---|---|
| 1 | Heat transfer-related variables | | SFN | continuous | / | The average annual value of SNF from 2014 to 2019 | Shan et al. [43] | 1km |
| 2 | | | ASCD | continuous | Days | The average ASCD values for the 10 years prior to the EGIR reporting time | MOD10A1.006 | 500m |
| 3 | Vegetation-related variables | | NDVImax | continuous | / | The maximum NDVI for the three years prior to the EGIR reporting time | MOD13Q1.061 | 250m |
| 4 | | | NDVImean | continuous | / | The mean NDVI for the three years prior to the EGIR reporting time | | 250m |
| 5 | | | VEG | categorical | / | Vegetation types | ncdc.ac.cn | 1km |
| 6 | Soil properties (0-200cm) | | SOC | continuous | dg/kg | Soil organic carbon (average values of 0-200cm) | SoilGrids250. (Hengl et al. [46]) | 250m |
| 7 | | | SCF | continuous | $cm^3/dm^3$ | Soil coarse fragment (average values of 0-200cm) | | 250m |
| 8 | | | SBD | continuous | $cg/cm^3$ | Soil bulk density (average values of 0-200cm) | | 250m |
| 9 | Topographic analysis variables | | DEM | continuous | m | Elevation | SRTM V3 DEM | 30m |
| 10 | | Moisture-related variables | DEV | continuous | / | DEV(R = 1km) | Calculated based on SRTM V3 DEM | 30m |
| 11 | | | TPI | continuous | / | TPI(R = 1km) | | 30m |
| 12 | | Terrain variables | slope | continuous | ° | Slope | | 30m |
| 13 | | | aspect | continuous | ° | Aspect | | 30m |
| 14 | | | RN | continuous | / | Surface roughness | | 30m |
| 15 | | | γ | continuous | / | Slope position | | 30m |
| 16 | | | LF | categorical | / | Landform | Theobald et al. [50] | 90m |
| 17 | | Ecologically-related variables | CHILI | continuous | / | Continuous Heat-Insolation Load Index | | 90m |
| 18 | | | mTPI | continuous | / | Multi-Scale Topographic Position Index | | 270m |
| 19 | | | TD | continuous | / | Topographic Diversity | | 270m |

*Please refer to the references for detailed meanings of variable values, such as VEG and LF.

position of the central point ($z_0$) using TPI and the standard deviation of the elevation [47–49]:

$$TPI = z_0 - \frac{1}{n_R}\sum_{i \in R} z_i \tag{7}$$

$$DEV = TPI/\sqrt{\frac{1}{n_R - 1}\sum_{i=1}(z_i - \bar{z})^2} \tag{8}$$

Both TPI and DEV can characterize the relative degree of water content in space, therefore they are used to evaluate relative soil moisture. The results of the factor detector with different $R$ were compared, and it was found that the $q$ value of TPI and DEV with $R$ taken as 1km was generally higher than the others (100m, 300m, 2km) in all highways, so the calculation $R$ of TPI and DEV was taken as 1km.

Terrain variables such as slope, aspect, elevation, and surface roughness (RN), can be extracted by utilizing DEM based on conventional terrain analysis algorithms. Slope position (γ) is also considered as a terrain variable in this study. It is defined as the relative position of a

point on the slope on which it is located and is calculated as:

$$\gamma = 100\% \cdot (e - e_{min})/(e_{max} - e_{min}) \tag{9}$$

Where $e$ is the elevation of the point, the $e_{max}$ and $e_{min}$ denote the elevation at the top and foot of the slope where the point is located, respectively.

Some ecologically-related variables are also considered as explanatory variables. The Continuous Heat-Insolation Load Index (CHILI) serves as a proxy for assessing the combined impact of insolation and topographic shading on evapotranspiration. It quantifies this effect by calculating the insolation received during the early afternoon when the sun's altitude is equivalent to that during the equinox. Multi-Scale Topographic Position Index (mTPI) is a dataset obtained by calculating TPI with different R in different scenarios. mTPI uses moving windows with $R$ (km) of 115.8, 89.9, 35.5, 13.1, 5.6, 2.8, and 1.2. By combining the CHILI and the mTPI datasets, Landform (LF) dataset was created.

Topographic diversity (TD) is a surrogate variable that represents the variety of temperature and moisture conditions available to species as local habitats. It expresses the logic that a higher variety of topo-climate niches should support higher diversity (especially plant) and support species persistence given climatic change.TD is calculated as:

$$TD = 1 - ((1 - mTPI) \cdot (1 - CHILI)) \tag{10}$$

The CHILI, mTPI, LF, and TD datasets are provided by the Conservation Science Partners Ecologically Relevant Geomorphology Datasets (accessed on 1 April 2023), for more detailed information please refer to Theobald et al. [50].

Table 3 lists more information about each explanatory variable. In this study, the spatial scale of OPGD was selected as the highest resolution in the explanatory variables, i.e., 30m.

## 3. Results

The OPGD model was applied in the analysis for permafrost distribution data of each highway (WX, GM, JC, KG, SL, and YC) and all highways together (TG). All factor detector and interaction detector results have passed the significance test of 99.9% ($p<0.001$).

The factor detection results of each highway are displayed in S1 Table. Overall, most variables show interaction enhancement with others; heat transfer-related variables and terrain variables generally contribute more to permafrost distribution compared with other variables; vegetation-related variables and moisture-related variables have smaller $q$ values and relatively lower contribution than other variables.

The variables with the highest contribution in factor detector results and interaction detector results were listed for each highway (Table 4). Several variables with high contribution to permafrost distribution were selected as the H-factors: γ, SFN, DEM, TD, and ASCD. These variables have the highest contribution in at least one highway in factor detector results. Also, since DEM has significant interactive contribution with other variables in interaction detection results of many highways, DEM is added as an H-factor. Table 5 and Fig 2 show the $q$ values and rank by the factor detector results of H-factors in each highway.

Results of factor detector show that γ is the major contributor to permafrost distribution. γ contributes most to the permafrost distribution in WX, GM, SL, and TG highways, with $q$ values of 0.1662, 0.115, 0.1231, and 0.1104, respectively. The major contributor to permafrost distribution is ASCD for YC highway ($q = 0.415$), and SFN and TD for JC and KG highways, with $q$ values of 0.1644 and 0.2643, respectively. The secondary variables varied among the highways are SFN (WX highway), DEV (GM highway), SCF (JC highway), CHILI (KG highway), VEG (SL highway), and DEM (YC highway), respectively.

**Table 4. Factor detector and interaction detector results.**

| Highway | Factor detector | | | | Interaction detector | | |
|---|---|---|---|---|---|---|---|
| | Major contributor | q-value | Secondary contributor | q-value | Major interaction | q-value | Interaction type |
| WX | γ | 0.1662 | SFN | 0.1042 | SFN∩TD | 0.2964 | Non-linear enhancement |
| GM | γ | 0.1150 | DEV | 0.0632 | SFN∩DEM | 0.2392 | |
| JC | SFN | 0.1644 | SCF | 0.1257 | SFN∩TD | 0.5117 | |
| KG | TD | 0.2643 | CHILI | 0.2033 | SFN∩SOC | 0.4808 | |
| SL | γ | 0.1231 | VEG | 0.0661 | SFN∩TD | 0.3242 | |
| YC | ASCD | 0.4158 | DEM | 0.3478 | DEM∩ASCD | 0.7170 | |
| TG | γ | 0.1104 | SFN | 0.0833 | SFN∩DEM | 0.1575 | |

The interaction detector reveals the impacts of interactions of variables (Table 4), where the interactions between SFN and other variables have the highest contribution for all highways except the YC highway. SFN∩TD is found to be the strongest interaction on WX ($q = 0.298$), SL ($q = 0.324$), and JC ($q = 0.493$) highways. The strongest interaction of GM and TG is SFN∩DEM; the strongest interaction of YC is DEM∩ASCD. The results of factor detector and interaction detector of H-factors are shown in Fig 3, and most of the H-factors interactions show enhancement for all highways. Among them, the impacts of interactions of SFN, TD, and DEM contribute the most, and the impacts of these three variables are enhanced by each other. Especially, SFN, whose $q$ values are always higher when interacting with TD, DEM, ASCD, and SOC, suggests that SFN plays an essential role in the spatial distribution of permafrost. Interestingly, the contribution of γ is weakened in the H-factors interaction, although γ is the major contributor in the results of factor detector.

## 4. Discussion

### 4.1 Indicators of permafrost

Identifying appropriate indicators to characterize the likelihood of permafrost distribution is crucial, given that the distribution pattern plays a vital role in engineering and environmental practices. Fig 4 shows the results of risk detector for permafrost distribution in TG analysis, with red and blue highlighting the strata with the highest and lowest permafrost risk values.

It can be seen that TD and RN show a clear pattern of decreasing risk values with increasing variable values. The complex surface conditions (with high TD and RN values) are due to the occurrence of surface movement or settlement in history, which leads to an imbalance in the hydrothermal state below the surface. At the same time, the uneven hydrothermal state can exacerbate surface movement or settlement. In this process, the degradation of permafrost is

**Table 5. q values and rank by the factor detector results of H-factors.**

| Highway | q values and rank of H-factors | | | | | | | | | |
|---|---|---|---|---|---|---|---|---|---|---|
| | γ | No. | SFN | No. | DEM | No. | TD | No. | ASCD | No. |
| WX | 0.1662 | 1 | 0.1042 | 2 | 0.0819 | 5 | 0.0698 | 7 | 0.0994 | 3 |
| GM | 0.1150 | 1 | 0.0586 | 4 | 0.0534 | 5 | 0.0198 | 15 | 0.0380 | 6 |
| JC | 0.0995 | 7 | 0.1644 | 1 | 0.1213 | 3 | 0.1168 | 4 | 0.1164 | 5 |
| KG | 0.106 | 11 | 0.1402 | 5 | 0.0862 | 16 | 0.2643 | 1 | 0.1202 | 9 |
| SL | 0.1231 | 1 | 0.0185 | 11 | 0.0474 | 3 | 0.0465 | 4 | 0.0290 | 7 |
| YC | 0.1326 | 5 | 0.0889 | 6 | 0.3478 | 2 | 0.0683 | 8 | 0.4158 | 1 |
| TG | 0.1104 | 1 | 0.0833 | 2 | 0.0317 | 7 | 0.0163 | 12 | 0.0145 | 15 |

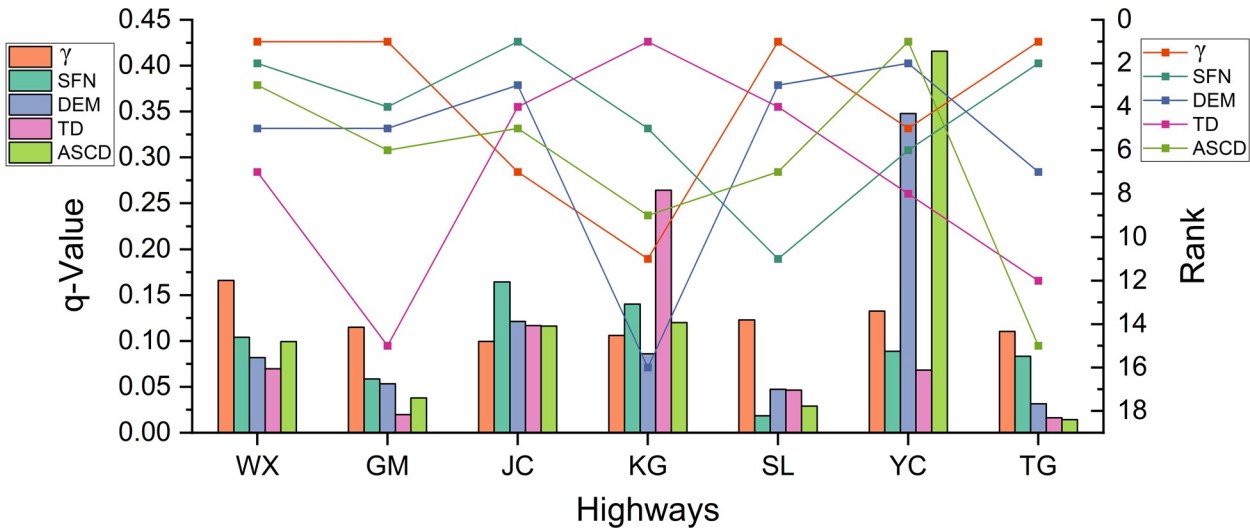

**Fig 2. *q* values and rank by the factor detector results of H-factors.**

both the cause and result of both [51]. In contrast, in areas with lower surface complexity, the hydrothermal state below the surface is relatively balanced, and the permafrost state is relatively stable.

The distribution of risk values of NDVImax and NDVImean show low at both ends and high in the middle, which indicates that either too dense or too limited vegetation is not conducive to permafrost. The main vegetation types in the study area are forests, shrubs, and marshes, with relatively high vegetation coverage. The climate change-induced rise in vegetation activity and prolonged growing seasons can exacerbate regional warming in this area, impacting permafrost preservation [52, 53]. Vegetation plays a role in intercepting snow, redistributing solar radiation, and dense vegetation can effectively isolate the influence of permafrost and surface thermal effects on the underlying surface, promoting the development and protection of permafrost [54]. In addition, permafrost degradation can alter the thermal state of the soil, prompting a transition in surface vegetation types from coniferous forests to broad-leaved forests [55]. Meanwhile, degraded permafrost provides moisture supply to vegetation, leading to an increase in vegetation coverage [56].

Additionally, we observed that the risk value is highest when SOC is in the range (617,724] and SCF is in the range (254,267]. SBD show a clear pattern of decreasing risk values with increasing variable values. The ability of permafrost to conduct and retain heat depends on the soil's thermal conductivity and heat capacity. Developed peat and muck can ensure the enrichment of surface moisture, thereby directly influencing the abundance of underground ice [57].

SFN shows a pattern of increasing risk values with increasing variable values. SFN is used to represent and categorize the stability of permafrost. When SFN is greater than 0.5, it indicates the presence of permafrost, and as SFN increases, the continuity of permafrost also rises [42]. The results of risk detector validate the feasibility of SFN in indicating permafrost at the field scale. When ASCD is relatively small, the risk value is higher. In cases of thin snow cover, soil temperature is more susceptible to air temperature influence, especially during the impact of cold air, leading to rapid development of permafrost. However, when snow cover is thick, the insulating effect of snow on ground temperature also provides insulation for the permafrost. When the soil temperature exceeds a certain threshold above the air temperature, shallow

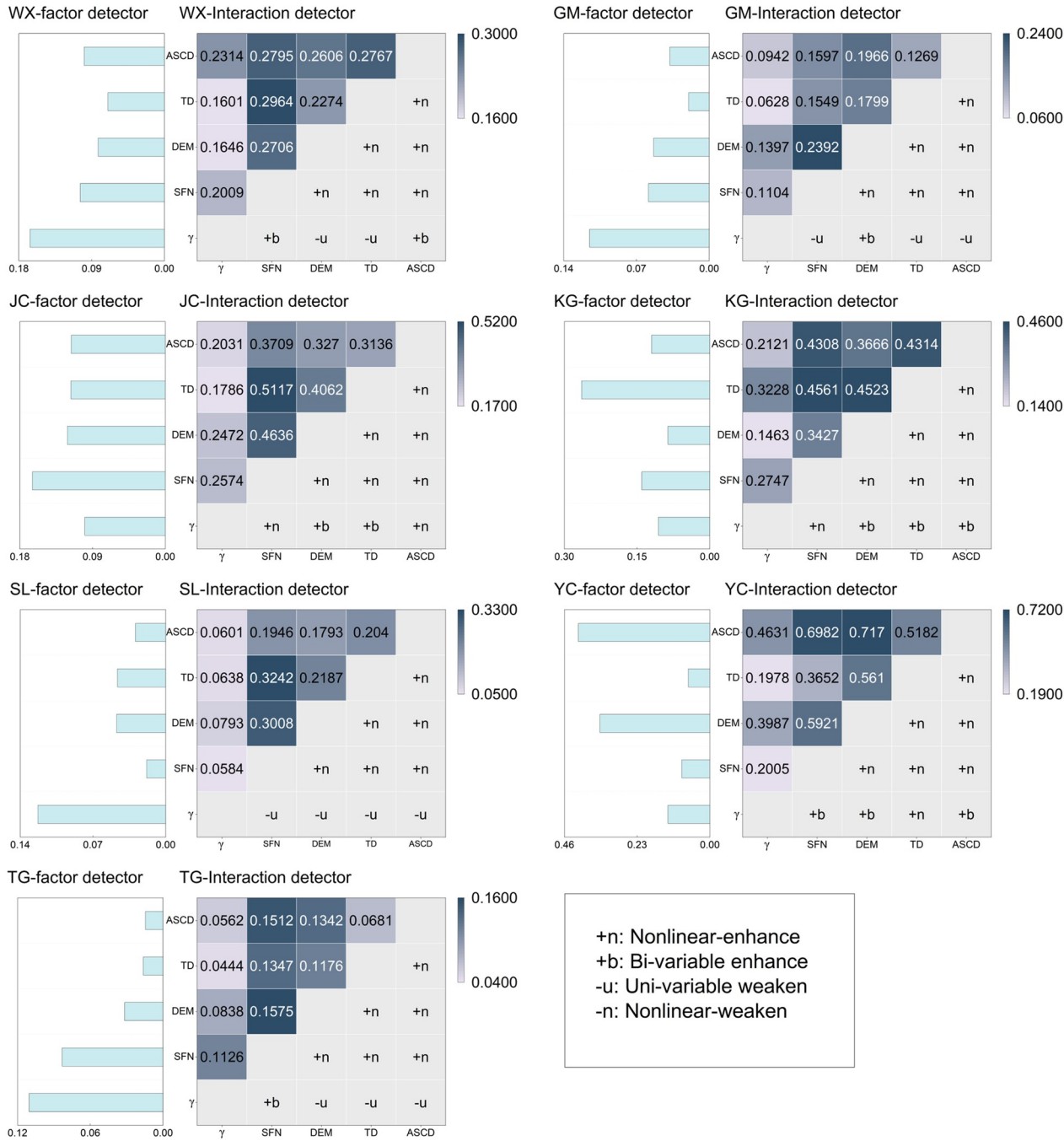

**Fig 3. Results of H-factors factor detection and interaction detection for each highway.**

permafrost may disappear due to the melting of water in soil pores, resulting in a reduction in permafrost extent [58].

In order to further explore the capability for variables to indicate permafrost, the ROC test was applied for analyzing the performance of variables with continuous values to indicate permafrost in TG (Fig 5).

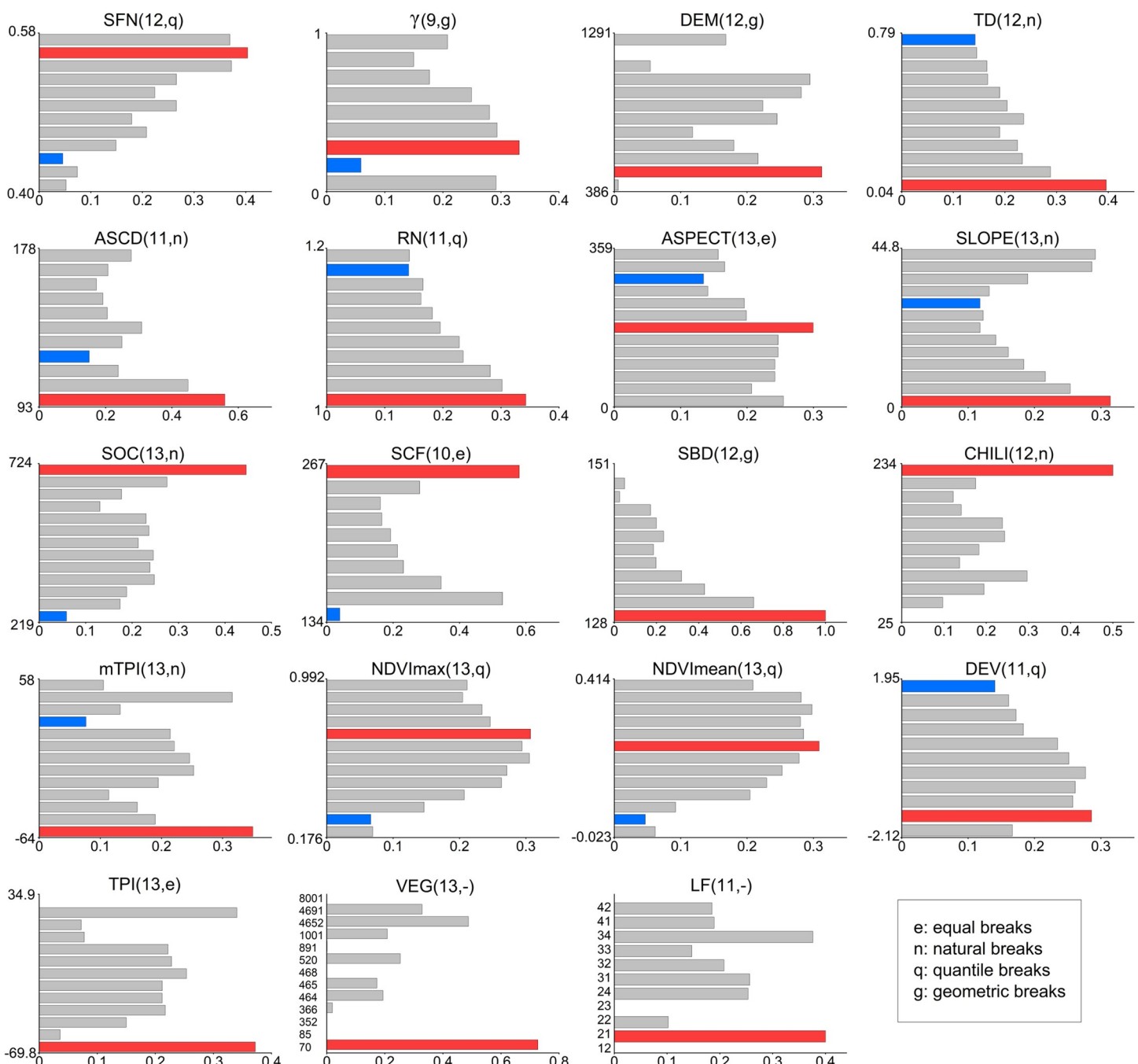

**Fig 4. Risk detection results for each variable in the analysis of permafrost distribution in TG.** Red and blue are used to highlight the strata with the highest and lowest risk values, respectively. In parentheses after the variables, the numbers indicate the number of strata for the variables, and the letters indicate the discretization method.

At the field scale, if a indication for permafrost area in numerical form is needed, SFN would be a better choice. Among all the variables, SFN had the highest AUC value of 0.7063. The AUC values of RN, slope, and NDVImean are higher than 0.6, while the AUC values of the remaining variables lie between 0.5 and 0.6. The spatial resolution of SFN used in this study was 1 km, but SFN had the strongest performance to indicate permafrost among all the variables, which verifies the feasibility of using SFN at the field scale to distinguish between

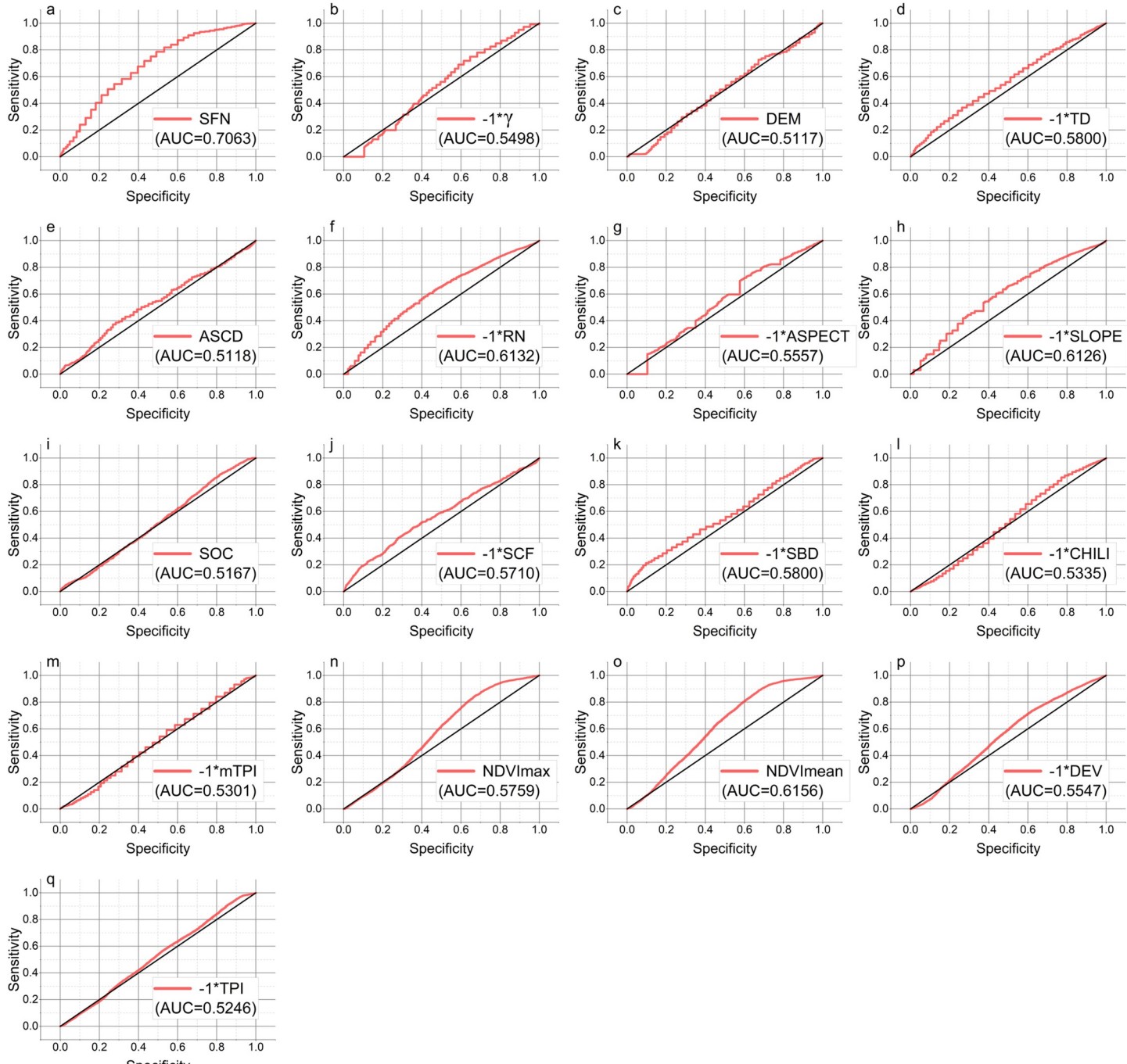

**Fig 5. ROC test curves and AUC values for each variable on the distribution of permafrost in TG.**

permafrost and non-permafrost areas. Further, downscaling SFN may be a possible solution for high-resolution permafrost mapping.

Although the AUC values of many variables are not high, the results of the risk detector exhibit a certain distribution pattern. This suggests that the permafrost probability does not increase or decrease monotonically with the increase of these variables and that there may be a mathematical relationship between the permafrost probability and the values of the variables.

Therefore, it is speculated that some mathematical deformation between variables would enhance their AUC values, i.e., their performance to indicate permafrost. This part of the study will carry on in the following work.

## 4.2 γ, a special explanatory variable for permafrost

Results of factor detector show that the $q$ values of γ are relatively stable across highways, lying between 0.1 and 0.2. The explanatory variables with the highest contribution to permafrost distribution in the WX, GM, SL, and TG highways is γ (Table 1). This suggests that the distribution of permafrost at the field scale is closely related to γ in NEC. The topography of the permafrost region in NEC is complex, with numerous intricate mountains and rivers, a humid climate, and extensive forest, making it easy to form microclimate zones [59]. On a slope surface, different slope positions usually have different vegetation cover, which affects the heat transfer between the ground and the air [60]. In addition, the soil environment is usually different on different slope positions, for example, the soil is wetter on the lower part, and this can also have an impact on the permafrost.

In the WX, GM, JC, KG highways, and TG analysis, the risk value tends to decrease with the increase of γ value (Fig 6). The distribution of permafrost at the field scale confirms the "Xingan-Baikal type" permafrost degradation pattern: degradation occurs first at high places and later at low places. The distribution of permafrost in the study area is strongly influenced by terrain factors, and is mostly developed and distributed between valley bottoms, depressions, low terraces, and (semi) sunny slopes [61]. Lowlands favor the development and protection of permafrost. Especially in the paludified areas with mosses and peat layers in basins, due to the influence of soil type, moisture content, and continuous winter air stagnation in mountainous lowlands, permafrost is more developed and has a higher ice content [62]. In addition, the risk values of slope are high at both ends and low in the middle (Fig 5) which indicates that permafrost is more likely to present in places with relatively steep or relatively gentle slopes also confirms the terrain distribution pattern of permafrost.

On the other hand, the mean risk value of each interval of γ is smaller for SL and YC highways when compared with other highways, and the risk value does not decrease with the increase of γ value. SL and YC highways are located at the boundary of permafrost region (Fig 1B), where permafrost degradation is serious and the proportion of permafrost sections to the total length of the highways is small. Severe permafrost degradation has resulted in ground conditions and climate scenarios become the major sources of uncertainty for high-resolution permafrost study [63].

In the interaction detector analysis, most interactions between γ and other H-factors show a uni-variable weakening, a few show a bi-variable enhancement, and very few show a nonlinear enhancement (Fig 3). In most cases, the $q$ value of γ after interacting with other variables is higher than that of the variable alone, but lower or slightly higher than that of γ alone. This is because the boundaries of slopes are very clear when calculating γ, resulting in γ not being smoothly continuous in space, which is different from the other variables. Therefore, γ is considered a strong but independently applicable explanatory variable for permafrost.

## 4.3 The differences in the manifestation of permafrost along different highways

The six highways are situated in different spatial geographical locations (Fig 1 and Table 1). WX is the highway with the highest latitude among the six, and it runs predominantly in an east-west direction. Therefore, WX can avoid the impacts caused by changes in latitude well. The major contributor to the permafrost distribution of WX is γ. In the WX highway, along

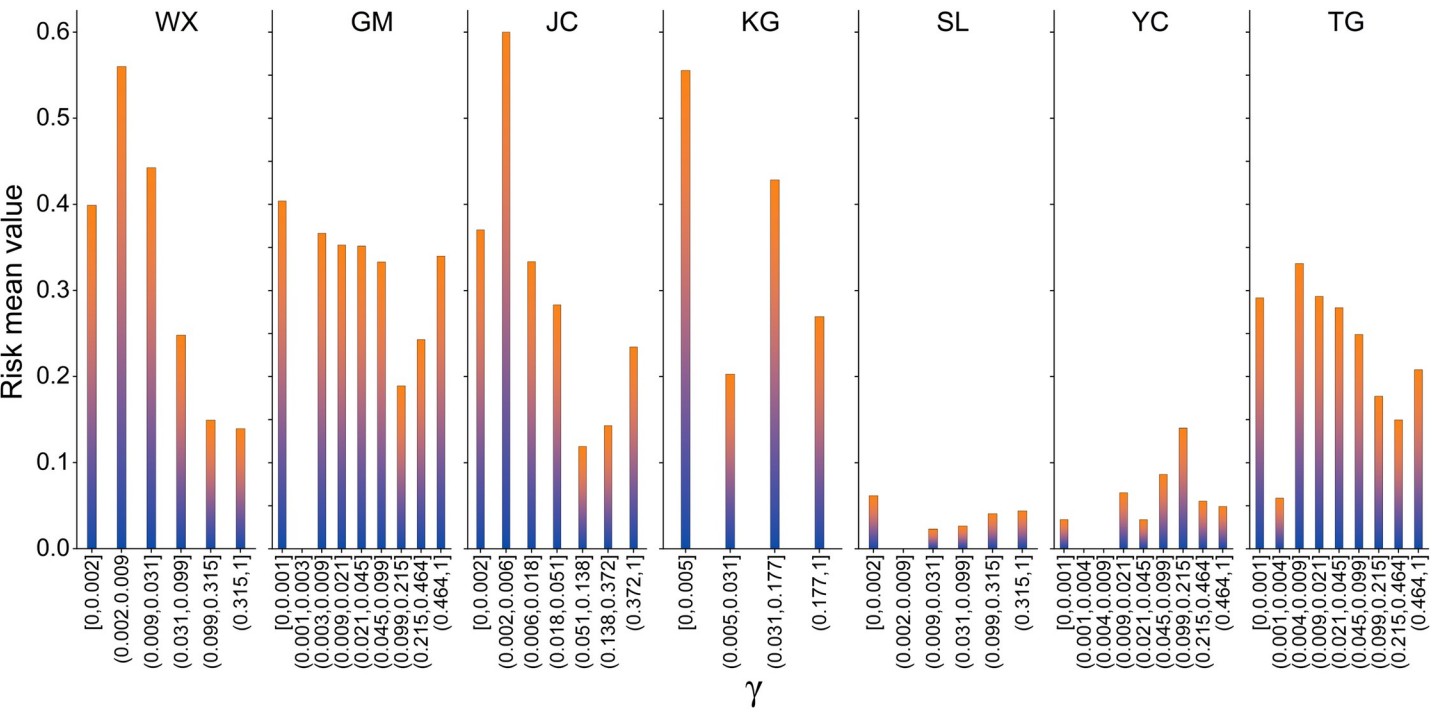

**Fig 6. Risk detector results of γ in each highway.**

the highway route, permafrost sections start with decreasing γ values and end with increasing γ values (Fig 7A1 and 7A2). For areas with generally high γ values, small areas of low γ values may also reserve permafrost (Fig 7A3). Similar phenomena occur on the GM and SL highways, in which γ is also the major contributor. This may be associated with the relatively large spatial extent of the three highways.

KG's latitude and longitude are in a central position. KG is located in the central part of the Da Xing'anling Mountains, where the permafrost temperature is the lowest (Fig 1B), and the permafrost here is of the continuous type [21]. The proportion of the total length of the permafrost section is the highest at 57.94%. The major contributor to the permafrost distribution of KG is TD. In the KG highway, the permafrost sections distribute in areas with low TD values in overall highway (Fig 7B), which also shows locally (Fig 7B1). As discussed in Section 4.1, there is a possible correlation between order permafrost degradation order and TD values that is permafrost degrade first where TD is relatively high (Fig 7B2 and 7B3). In continuous permafrost regions, the correlation is even stronger, making TD one of the main factors affecting permafrost degradation.

JC is the highway with the smallest latitude and longitude range among the six, and it is characterized by simple terrain conditions, located within a single hydrological unit. In such a homogeneous environmental setting with relatively stable conditions, SFN is the major contributor to permafrost distribution. In the JC highway, SFN indicates the distribution of permafrost sections in general, and permafrost sections are mainly distributed in areas with high SFN values (Fig 7C). The principle of SFN is to reflect the thermal state of the ground by directly calculating the freezing and thawing status of the surface, therefore, although there is a limitation by the resolution of SFN (1 km), areas with high concentrations of SFN values tend to have more continuous permafrost (Fig 7C3).

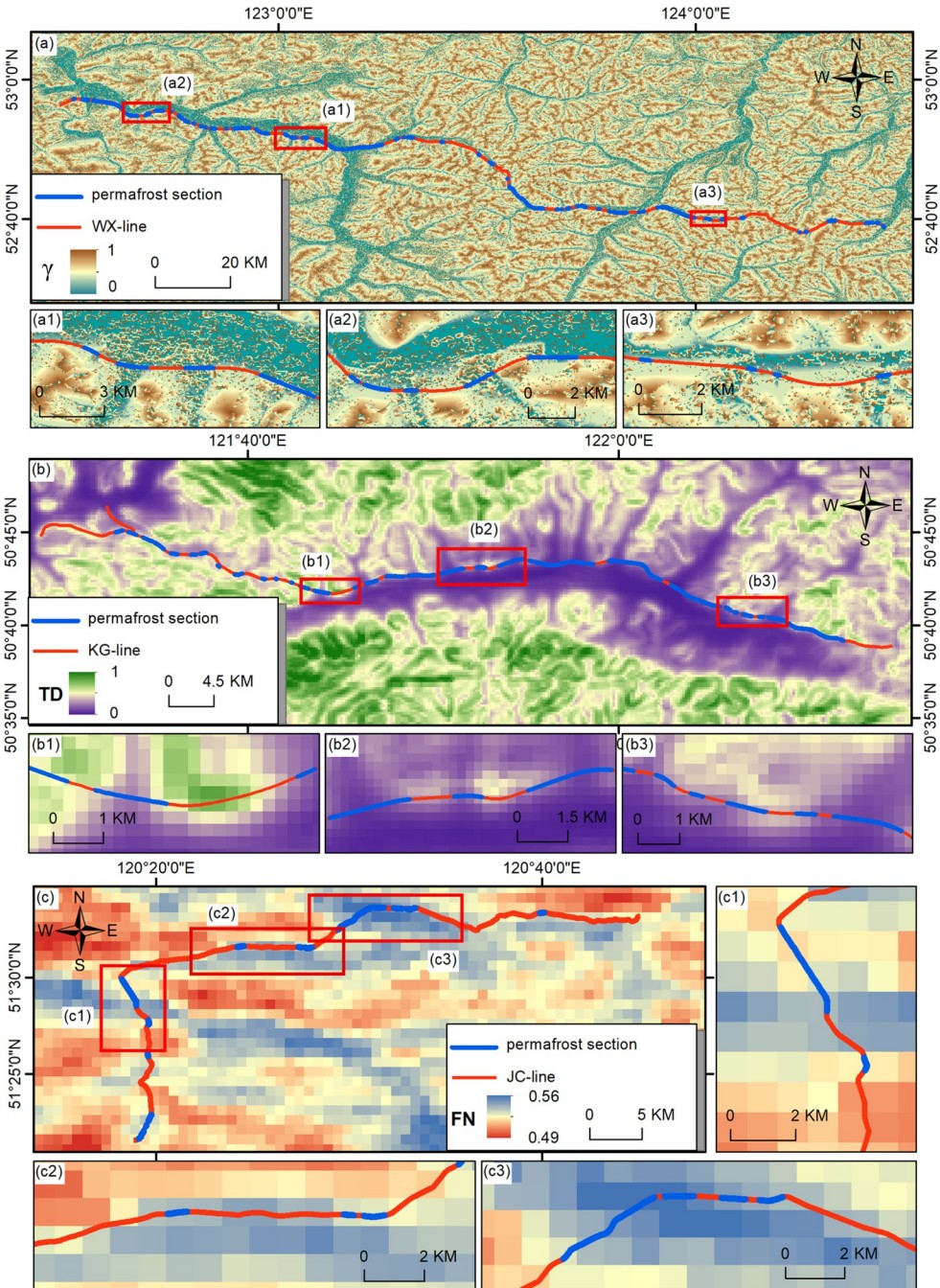

**Fig 7.** Spatial distribution of permafrost sections in WX (a), KG (b), JC (c) highways and the values of γ, TD, SFN, respectively. γ data was calculated through elevation data provided by NASA Land Processes Distributed Active Archive Center's free 30 m SRTM DEM ((SRTM V3, https://lpdaac.usgs.gov/, (accessed on 1 April 2023))). TD data was provided by the Conservation Science Partners Ecologically Relevant Geomorphology Datasets (accessed on 1 April 2023). SFN data is provided by Shan et al. [43].

YC is located at the southern end of the Greater Khingan Mountains, being the southern-most and highest-altitude highway among the six (Table 1). This area marks the southern boundary of permafrost distribution in the NEC, and permafrost distribution here continues

to degrade [64]. Unlike other highways, the major contributor for YC is ASCD, followed by DEM, with their q-values being close. The interactions between DEM and ASCD have the highest contribution. Snow cover and altitude are the main factors controlling permafrost distribution in this area, where the permafrost exhibits some characteristics of mountain permafrost, with more permafrost located at higher elevations than at lower elevations [34]. In this area, the presence of a temperature reversal layer and hot springs is relatively common, resulting in warmer and more humid conditions in the lower parts, while the higher parts are relatively dry and cold.

## 4.4 Uncertainty

It must be admitted that this study still has some uncertainties and limitations. Firstly, the influence of clouds and the atmosphere may lead to uncertainties in the SFN, ASCD and NDVI data for this work. Therefore, it is still needed to further verify the results by using higher resolution and higher quality SFN, ASCD and vegetation index datasets. In addition, the samples for this study are derived from highways, and their spatial distribution is linear. Whenever conditions allow, it is advisable to use samples that are more random and have a spatial distribution across an area rather than along a linear path in future research.

## 5. Conclusion

In this study, the spatial stratified heterogeneity analysis of permafrost distribution at the field scale was applied based on the EGIR of six highways in the permafrost region of NEC. In addition, the performance of environmental variables in indicating permafrost is discussed. The conclusions are summarized as follows:

1. H-factors: γ, SFN, DEM, TD, and ASCD are the major contributors to the distribution of permafrost in NEC at the field scale. Among them, γ has the highest contribution than other variables. The risk detector results show that, in most cases, permafrost risk values show a decreasing trend with increasing γ values.

2. Interactions can enhance the explanatory variables' impact on permafrost distribution, especially the interactions between H-factors. Interaction between SFN and other variables (especially TD and DEM) always has relatively high $q$ values. However, γ is always weakened by interaction with other variables.

3. According to the results of risk detector, TD, RN, and SBD show a pattern of decreasing permafrost risk values with increasing variable values; SFN shows a pattern of increasing permafrost risk values with increasing variable values. The risk of permafrost is significantly high when SOC $\in$ (617,724], SCF $\in$ (254,267], and CHILI $\in$ (216,234]. Results of the ROC test show that, among all the variables, SFN (AUC = 0.7063) has the best indication performance for permafrost distribution at the field scale.

4. There is spatial heterogeneity in the distribution of permafrost on highways in different spatial locations. γ is the major contributor to permafrost distribution in WX, GM and SL. TD, SFN and ASCD are the major contributors in KG, JC and YC respectively. Permafrost sections of highways are correlated with spatial relative values of environmental variables.

## Supporting information

**S1 Table. *q* values of variables by factor detector in each highway.**
(DOCX)

## Author Contributions

**Conceptualization:** Ying Guo, Lisha Qiu, Wei Shan.

**Data curation:** Shuai Liu, Lisha Qiu, Chengcheng Zhang, Wei Shan.

**Formal analysis:** Ying Guo, Shuai Liu.

**Funding acquisition:** Ying Guo, Wei Shan.

**Investigation:** Ying Guo, Wei Shan.

**Methodology:** Shuai Liu, Lisha Qiu.

**Project administration:** Ying Guo, Chengcheng Zhang, Wei Shan.

**Resources:** Shuai Liu, Chengcheng Zhang, Wei Shan.

**Software:** Shuai Liu, Lisha Qiu.

**Supervision:** Ying Guo.

**Validation:** Shuai Liu, Lisha Qiu.

**Visualization:** Shuai Liu, Lisha Qiu.

**Writing – original draft:** Shuai Liu, Wei Shan.

**Writing – review & editing:** Ying Guo, Wei Shan.

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
