## [Decision Letter · Decision Letter 0]

23 Oct 2023

PONE-D-23-25976Spatial stratified heterogeneity analysis of field scale permafrost in Northeast China based on optimal parameters-based geographical detectorPLOS ONE

Dear Dr. Shan,

Thank you for submitting your manuscript to PLOS ONE. After careful consideration, we feel that it has merit but does not fully meet PLOS ONE’s publication criteria as it currently stands. Therefore, we invite you to submit a revised version of the manuscript that addresses the points raised during the review process.

We look forward to receiving your revised manuscript.

Kind regards,

Sher Muhammad, PhD

Academic Editor

PLOS ONE

Journal Requirements:

"We thank the National Natural Science Foundation of China (Grant No. 41641024) and Science and the Technology Project of Heilongjiang Communications Investment Group (Grant No.JT-100000-ZC-FW-2021-0182) for providing financial support and the Field scientific observation and research station of the Ministry of Education-Geological environment system of permafrost area in Northeast China (MEORS-PGSNEC)."

"We thank the National Natural Science Foundation of China (Grant No. 41641024) and Science and the Technology Project of Heilongjiang Communications Investment Group (Grant No.JT-100000-ZC-FW-2021-0182) for providing financial support and the Field scientific observation and research station of the Ministry of Education-Geological environment system of permafrost area in Northeast China (MEORS-PGSNEC)."

"We thank the National Natural Science Foundation of China (Grant No. 41641024) and Science and the Technology Project of Heilongjiang Communications Investment Group (Grant No.JT-100000-ZC-FW-2021-0182) for providing financial support and the Field scientific observation and research station of the Ministry of Education-Geological environment system of permafrost area in Northeast China (MEORS-PGSNEC)."

5. We note that [Figures 1 and 6] in your submission contain [map/satellite] images which may be copyrighted. All PLOS content is published under the Creative Commons Attribution License (CC BY 4.0), which means that the manuscript, images, and Supporting Information files will be freely available online, and any third party is permitted to access, download, copy, distribute, and use these materials in any way, even commercially, with proper attribution. For these reasons, we cannot publish previously copyrighted maps or satellite images created using proprietary data, such as Google software (Google Maps, Street View, and Earth). For more information, see our copyright guidelines: http://journals.plos.org/plosone/s/licenses-and-copyright.

a. You may seek permission from the original copyright holder of Figures 1 and 6 to publish the content specifically under the CC BY 4.0 license.  

Additional Editor Comments:

Two anonymous reviewers have evaluated the manuscript, both of them suggested major review. Both of them appreciated the study for its innovative approach in analyzing spatial stratified heterogeneity of permafrost distribution in Northeast China with valuable data sources. However, they pointed out several areas for improvement including restructuring the content for better clarity, emphasizing the significance of the study, enhancing the interpretation of risk detectors, addressing the effects of zonation, and considering practical significance when choosing segmentation points. They also recommended presenting q-values for all variables, refining expressions for clarity, and improving language for readability. Additionally, they suggested adding relevant data to demonstrate permafrost degradation severity, highlighting the study's significance in the introduction, providing clearer figure labels, explaining the detection of multiple ecological factors, considering temperature and precipitation in factor selection, comparing results with other studies, discussing limitations, and providing more mechanistic explanations in the discussion section. The authors are advised to improve the manuscript considering the reviewers comments and resubmit the revised version.

Reviewers' comments:

Reviewer's Responses to Questions

**Comments to the Author**

1. Is the manuscript technically sound, and do the data support the conclusions?

Reviewer #1: Yes

Reviewer #2: Partly

2. Has the statistical analysis been performed appropriately and rigorously? 

Reviewer #1: Yes

Reviewer #2: Yes

3. Have the authors made all data underlying the findings in their manuscript fully available?

Reviewer #1: Yes

Reviewer #2: No

4. Is the manuscript presented in an intelligible fashion and written in standard English?

Reviewer #1: Yes

Reviewer #2: Yes

5. Review Comments to the Author

Reviewer #1: Based on the OPGD and ROC test, this study analyzed the spatial stratified heterogeneity of permafrost distribution and the indicating performance of environmental variables on permafrost in NEC. The results of this study may help to further understand the relationship between climate change and permafrost degradation. However, there are some concerns that the authors should address before it can be considered for publication.

1. Lines 53-54，I suggest the authors add relevant data to demonstrate the severity of permafrost degradation.

2. In the last paragraph of the introduction, I suggest the authors further highlight the significance of this study.

3. To present the figures more clearly, the authors should change the latitude and longitude of Figure 1 to English expression.

4. In GD model, why didn't the authors add an introduction to the detection of multiple ecological factors?

5. Why didn't the author consider temperature and precipitation when selecting influencing factors?

6. In order to further highlight the innovation of this article, it is better to compare the results of this study with other studies.

7. A paragraph of limitation discussion should be added to clarify the limitation or uncertainty of current study. For example, the uncertainty of remote sensing data including NDVI data (e.g. Ma et al., 2022; Shen et al., 2022) may affect the research results.

8. In discussion, more mechanism explanations should be added to further explain the relationship between permafrost degradation and terrain, climate, and vegetation.

References:

Vegetation greening, extended growing seasons, and temperature feedbacks in warming temperate grasslands of China. Journal of Climate, 2022, 35(15): 5103-5117.

Variation of vegetation autumn phenology and its climatic drivers in temperate grasslands of China. International Journal of Applied Earth Observation and Geoinformation, 2022, 114: 103064.

Reviewer #2: The manuscript analyzes the spatial stratified heterogeneity of the permafrost distribution in Northeast China and explains the related causes with the field Engineering Geological Investigation Reports and related GIS/RS data products using the Geodetector method and the ROC detection method. Overall it is innovative and applicable, but it also has a bit of flaws, and the comments are as follows:

1. The content structure needs to be adjusted, e.g., the content of line358-366 should be adjusted to the methodology section.

2. Insufficient care and interpretation of the results of the risk detectors was carried out only on the magnitude of the resultant values. The risk detector focuses on detecting whether spatial patterns based on mean representations are significantly different between subregions categorized by categorical or hierarchical variables. This component needs to be strengthened.

3. Given the spatial heterogeneity, it is rather unfortunate that the authors did not address the effects of zonation, such as longitude and latitude, since there is a certain span of latitude and longitude for each highway.

4. When using the geodetector method, the pursuit of a large q-value is one of the goals, but the choice of segmentation points for variables needs to be based on practical significance and not on the size of the q-value. For example, the choice of R when calculating TPI and DEV (line224-278) needs to take into account the difference in actual spatial resolution, the difference in spatial resolution between the explanatory factor variables and the results based on the highway survey report.

5. As far as I know, the discretization method when optimizing a geodetector also includes the standard deviation (sd), why did the authors discard it. (line147-148).

6. In the results section it is recommended to give the results of the q-values (in the form of graphs or tables) for all variables for each highway, instead of the so-called highest degree of interpretation in the article. This is because the differences between the q-values are also important and need to be analyzed for this aspect of the content.

7. I am curious as to whether the authors used a certain range of mean values for the raster data extraction when pre-processing the geodetector input data, or were the values extracted directly by geographical location? The question still connotes the issue of spatial resolution differences between the data.

8. How to account for large differences in both time and space in the input variables in a way that does not lead to less feasible or even erroneous results.

9. Some expressions need to be adjusted, e.g., lines 136-138, ①, ②, ③, please delete; line 155, where a capital W is not required, same elsewhere; numbers one to nine without units should be presented as words; numbers 10 and over without units should be presented as numerals; and all numbers with units should be presented as numerals (e.g., line 174, same elsewhere).

10. It is recommended that the language of the article be polished to improve readability.

Some minor flaws and suggestions:

1. Line 21. The words “global climate change” suggest directly “global warming”.

2. Line 30. Suggests giving the definition of H-factor in the abstract section.

3. Line 46-47, give references for permafrost definition.

4. Line 192, “selection” should be “selecting”.

5. Line 252-253, table3, cm3/cm3 should be cm3/cm3. * Please change to Note: , same elsewhere.

6. Line 337, SLOPE, no need to capitalize, same elsewhere.

7. References, please adjust the formatting and writing style to be consistent, e.g. Line 431, Line 61.

8. Figures, there are Chinese characters in Figure 1. γ is written incorrectly in Fig. 2 and Fig. 6, if it is a size, it should be Γ.

6. PLOS authors have the option to publish the peer review history of their article (what does this mean?). If published, this will include your full peer review and any attached files.

Reviewer #1: No

Reviewer #2: No

---

## [Author Response · Author response to Decision Letter 0]

15 Dec 2023

Dear reviewers,

I wish to express my gratitude for your meticulous review and invaluable suggestions. We deeply appreciate the opportunity to learn from your extensive knowledge and experience. Your expertise and detailed evaluation have significantly contributed to the enhancement of our research. We have carefully considered and implemented each of your suggestions in the revised manuscript.

The highlighted portions indicate the changes made in the manuscript. The overall structure of the article has been adjusted, particularly in the discussion section where the content order has been rearranged, and additional material has been included. Additionally, to enhance the manuscript's conformity, some textual expressions have been adjusted without altering the original meaning; these sections are not specifically highlighted.

The revisions based on your suggestions have been thoroughly incorporated into the revised version, which can be found in the resubmitted manuscript. Thank you once again for your time and patience. We eagerly anticipate your feedback.

Best regards,

---

## [Decision Letter · Decision Letter 1]

27 Dec 2023

Spatial stratified heterogeneity analysis of field scale permafrost in Northeast China based on optimal parameters-based geographical detector

PONE-D-23-25976R1

Dear Dr. Shan,

We’re pleased to inform you that your manuscript has been judged scientifically suitable for publication and will be formally accepted for publication once it meets all outstanding technical requirements.

Kind regards,

Sher Muhammad, PhD

Academic Editor

PLOS ONE

Additional Editor Comments (optional):

The revision has improved the manuscript quality and recommended for publication.

Reviewers' comments:

Reviewer's Responses to Questions

**Comments to the Author**

1. If the authors have adequately addressed your comments raised in a previous round of review and you feel that this manuscript is now acceptable for publication, you may indicate that here to bypass the “Comments to the Author” section, enter your conflict of interest statement in the “Confidential to Editor” section, and submit your "Accept" recommendation.

Reviewer #1: All comments have been addressed

Reviewer #2: All comments have been addressed

2. Is the manuscript technically sound, and do the data support the conclusions?

Reviewer #1: Yes

Reviewer #2: Yes

3. Has the statistical analysis been performed appropriately and rigorously? 

Reviewer #1: Yes

Reviewer #2: Yes

4. Have the authors made all data underlying the findings in their manuscript fully available?

Reviewer #1: Yes

Reviewer #2: Yes

5. Is the manuscript presented in an intelligible fashion and written in standard English?

Reviewer #1: Yes

Reviewer #2: Yes

6. Review Comments to the Author

Reviewer #1: (No Response)

Reviewer #2: I thank the authors for adopting the review comments and revising the manuscript. Compared with the first version, this version has made great progress, both in content and structure. I think the revised version is ready for publication, when some of the more detailed changes are completed, such as the extra punctuation mark "." in the title of section 4.3.

7. PLOS authors have the option to publish the peer review history of their article (what does this mean?). If published, this will include your full peer review and any attached files.

Reviewer #1: No

Reviewer #2: No

---

## [Editor Report · Acceptance letter]

6 Feb 2024

PONE-D-23-25976R1 

PLOS ONE

Dear Dr. Shan, 

I'm pleased to inform you that your manuscript has been deemed suitable for publication in PLOS ONE. Congratulations! Your manuscript is now being handed over to our production team.

Kind regards, 

on behalf of

Dr. Sher Muhammad 

Academic Editor

PLOS ONE